# HIV-1 Infection of Long-Lived Hematopoietic Precursors In Vitro and In Vivo

**DOI:** 10.3390/cells11192968

**Published:** 2022-09-23

**Authors:** Sebastian Renelt, Patrizia Schult-Dietrich, Hanna-Mari Baldauf, Stefan Stein, Gerrit Kann, Markus Bickel, Ulrikke Kielland-Kaisen, Halvard Bonig, Rolf Marschalek, Michael A. Rieger, Ursula Dietrich, Ralf Duerr

**Affiliations:** 1Georg-Speyer-Haus, Institute for Tumor Biology and Experimental Therapy, 60596 Frankfurt, Germany; 2Max von Pettenkofer Institute & Gene Center, Virology, National Reference Center for Retroviruses, Faculty of Medicine, LMU München, 81377 Munich, Germany; 3Institute of Medical Virology, Goethe University, 60596 Frankfurt, Germany; 4Department of Medicine II/Infectious Diseases, Goethe University Hospital, 60596 Frankfurt, Germany; 5Infektiologikum, Center for Infectious Diseases, 60596 Frankfurt, Germany; 6Department of Gynecology, Medical School, Goethe University, 60590 Frankfurt, Germany; 7Institute for Transfusion Medicine and Immunohematology, German Red Cross Blood Donor Service Baden-Württemberg-Hessen, Goethe University, 60528 Frankfurt, Germany; 8Institute of Pharmaceutical Biology, Goethe University, 60438 Frankfurt, Germany; 9Department of Medicine, Hematology/Oncology, Goethe University Hospital, 60590 Frankfurt, Germany; 10German Cancer Consortium (DKTK), German Cancer Research Center, 69120 Heidelberg, Germany; 11Frankfurt Cancer Institute, 60596 Frankfurt, Germany; 12Cardio-Pulmonary Institute, 60596 Frankfurt, Germany; 13Department of Microbiology, New York University Grossman School of Medicine, New York, NY 10016, USA

**Keywords:** HIV-1 infection, hematopoietic stem and progenitor cells (HSPCs), hematopoietic stem cells (HSC), multipotent progenitor cells (MPP), in vitro and in vivo, viral reservoir, proviral DNA, CXCR4 and CCR5 tropism, colony formation, replating assays

## Abstract

Latent reservoirs in human-immunodeficiency-virus-1 (HIV-1)-infected individuals represent a major obstacle in finding a cure for HIV-1. Hematopoietic stem and progenitor cells (HSPCs) have been described as potential HIV-1 targets, but their roles as HIV-1 reservoirs remain controversial. Here we provide additional evidence for the susceptibility of several distinct HSPC subpopulations to HIV-1 infection in vitro and in vivo. In vitro infection experiments of HSPCs were performed with different HIV-1 Env-pseudotyped lentiviral particles and with replication-competent HIV-1. Low-level infection/transduction of HSPCs, including hematopoietic stem cells (HSCs) and multipotent progenitors (MPP), was observed, preferentially via CXCR4, but also via CCR5-mediated entry. Multi-lineage colony formation in methylcellulose assays and repetitive replating of transduced cells provided functional proof of susceptibility of primitive HSPCs to HIV-1 infection. Further, the access to bone marrow samples from HIV-positive individuals facilitated the detection of HIV-1 *gag* cDNA copies in CD34+ cells from eight (out of eleven) individuals, with at least six of them infected with CCR5-tropic HIV-1 strains. In summary, our data confirm that primitive HSPC subpopulations are susceptible to CXCR4- and CCR5-mediated HIV-1 infection in vitro and in vivo, which qualifies these cells to contribute to the HIV-1 reservoir in patients.

## 1. Introduction

Although combination antiretroviral therapy (cART) can effectively suppress viral replication, HIV-1 persists in infected individuals for years, and viral rebounds occur upon treatment interruptions [1]. This is due to the ability of HIV-1 to establish productive as well as latent infection of cells [2]. The latently integrated proviral genome is not affected by cART because of halted replication and virion production. Therefore, neither conventional antiviral therapy nor protective immune responses can affect the proviral stage of the virus significantly [3,4]. In the absence of cART, HIV-1 provirus can be reactivated by cellular factors and metabolic changes, resulting in consecutive spreading of HIV-1 to non-infected cells [5,6,7]. Therefore, the characterization and eradication of latent viral reservoirs are crucial in finding a cure for HIV-1 [8,9,10,11,12,13,14,15,16,17,18,19,20,21,22,23,24].

Viral reservoirs are established early in HIV-1 infection and can be detected within the first three days after viral transmission [5,21]. Cell populations suitable to serve as viral reservoirs are susceptible to HIV-1 infection and have a long lifespan. Such cells have the ability to proliferate and carry over the latently integrated provirus to their progeny [8]. HIV-1 can persist in such cells as integrated provirus for time spans exceeding seven years [4,25,26,27]. Strategies to eradicate these latent reservoirs include “shock and kill” approaches: latency-reversing agents (LRAs) are used to activate viral replication (shock) to facilitate susceptibility of infected cells to antiviral cytotoxic immune mechanisms (kill), while simultaneous cART prevents spreading of infection to uninfected cells [28,29,30].

HIV-1 infects cells via interaction of the viral envelope (Env) protein with the cellular CD4 receptor. Additionally, co-ligation of a chemokine receptor, CCR5 or CXCR4, is needed to establish successful infection [31,32,33,34]. Early stages of infection are dominated by CCR5-tropic (R5) strains, whereas CXCR4-tropic (X4) strains are found predominantly in later stages of the disease. Infection with X4 HIV-1 has been reported to be associated with a more rapid rate of disease progression [35,36,37,38]. A shift from R5 to X4 tropism can be conferred by only a few defined amino acid changes within the V3 region of Env that can be acquired over time or due to outgrowth of a pre-existing minority population. X4 tropism comes with a more basic charge of V3, allowing genotypic predictions of co-receptor usage [39,40,41,42,43,44].

HIV-1 replicates preferentially in activated CD4^+^ T cells and macrophages [45,46]. In contrast, latent HIV-1 primarily persists in memory CD4^+^ T cells, the best-characterized viral reservoir. It has been shown that besides low-level viral replication and homeostatic proliferation of latently infected cells, clonal expansion of HIV-infected cells plays a primary role in maintaining the reservoir and ongoing HIV infection [47,48,49].

Within the CD4^+^ compartment, a cell population with stem-cell-like properties preferentially contributes to the viral reservoir [50]. Furthermore, HIV-1 sequences have been detected in HIV^+^ individuals that did not match those arising from the reservoir of resting memory CD4^+^ T cells, suggesting the existence of additional latent reservoirs [45,51,52]. These reservoirs have been discussed to involve not only different macrophage populations, such as intestinal or alveolar macrophages, but also dendritic cells and HSPCs [53,54,55,56,57,58,59,60].

Since hematopoietic defects and diseases are common in HIV-1 infection [61,62,63,64,65,66,67,68,69], the susceptibility of HSPCs to HIV-1 infection has been the subject of investigations for more than three decades [70,71,72,73,74,75,76,77,78,79,80,81,82]. Conceptually, HSPCs can serve as efficient HIV-1 reservoirs because they are long-lived, possess a broad variety of cell surface receptors, can proliferate and maintain self-renewal, and can differentiate into different cell types [83]. In recent years, there has been growing evidence of HIV-1 receptor expression on HSPCs [75,84,85,86,87,88,89,90] and their productive or latent HIV-1 transduction/infection in vitro and in vivo [74,75,76,79,80,91,92]. The Collins group found in an HIV-1 latency model that latent HIV-1 infection of HSPCs could be reversed by NF-κB activation [75,79]. While the infection of HSPCs has been controversial for many years, mainly due to insufficient cell purities in experimental settings [77,78], the preferential HIV-1 infection of highly purified CD4^high^ HSPC lineages (>99%) statistically excluded the possibility that cell contaminations accounted for the majority of the infectious events. Furthermore, identical HIV-1 sequences were detected in HSPCs and differentiated progenies (including CD4-negative cells that are highly resistant to HIV infection) over longer time in HIV-1-infected patients, suggesting that infected HSPCs contribute to latent reservoirs [93,94,95].

In this study, we add further evidence to the infection of highly marker-purified, naïve hematopoietic stem cells (HSCs) and multipotent progenitor cells (MPP) by HIV-1 in vitro and in vivo. Using HIV-1 reporter viruses, we could functionally demonstrate infection of HSPCs based on their capacity to form colonies persistently for up to 6 weeks post-infection. The analysis of hematopoietic progenitor cells from a limited number of bone marrow samples of HIV-1-infected patients indicated the presence of proviral HIV-1 DNA in hematopoietic progenitors in a substantial fraction of patient HSPCs, underlining their potential to constitute an in vivo HIV-1 reservoir.

## 2. Materials and Methods

### 2.1. Patient Samples

Whole umbilical cord blood was obtained pseudonymized from the Department of Gynecology, Goethe University Hospital, Frankfurt, Germany and the German Red Cross Blood Donation Service Baden-Württemberg—Hessen. Bone marrow samples of anonymous healthy donors were obtained from the German Red Cross Blood Donation Service Baden-Württemberg—Hessen from cell filter sets after bone marrow aspirations. Bone marrow aspirations of HIV-1-infected patients (pseudonymized) were obtained from the Department of Medicine II/Infectious Diseases, Goethe University Hospital, Frankfurt, Germany, during routine bone marrow analyses (ethical declarations can be found in the “Institutional Review Board Statement”).

### 2.2. Cell Isolation and Culture

Healthy donor bone marrow-derived mononuclear cells were washed out of bone marrow harvest collection kits. These kits consist of a combination of three sterile connected filters used to filter pieces of bones, large conglomerates of cells, and bone marrow stroma. The washout of the cells was performed using 25 mL cold RPMI1640 (Thermo Fisher Scientific, Waltham, MA, USA) supplemented with 1 mg collagenase II (100 units; Thermo Fisher Scientific) under constant, gentle shaking at room temperature followed by two washes of 50 mL cold 1 × PBS.

Bone marrow samples of people living with HIV (PLHIV) were taken during routine medical examinations and received in a sterile 10 mL syringe and purification of mononuclear cells was performed subsequently upon reception.

Mononuclear cells were purified using Ficoll-Hypaque centrifugation and frozen for long-term storage. Lin^−^ cells were isolated via two-round magnetic separation from Lin^+^ cells (CD2, CD3, CD11b, CD14, CD15, CD16, CD19, CD56, CD123, and CD235a) using the human Lineage Cell Depletion Kit according to the manufacturer’s instructions (Miltenyi Biotec, Bergisch-Gladbach, Germany). Lin^−^ cells were sorted into HSPC-subsets via flow cytometry using the antibodies indicated in Section 2.3, Antibodies used for flow cytometry. Isolated cells were cultured in SFEM-complete medium using StemSpan™ SFEM (STEMCELL Technologies, Vancouver, BC, Canada) supplemented with 100 ng/mL murine stem cell factor (mSCF), 100 ng/mL murine thrombopoietin (mTPO) (both cross-reactive with human cells), 10 ng/mL human IL-3, 10 ng/mL human FLT3 ligand (all PeproTech, Hamburg, Germany), 4 mM l-Glutamine, and (100 units/mL)/(100 µg/mL) penicillin/streptomycin (Pen/Strep, Thermo Fisher Scientific).

As a prerequisite of the study, a high purity of the studied HSPC populations was of central importance. While the possibility of contaminating CD3^+^ T cells cannot be fully ruled out, it was minimized with our sequential two-column lineage depletion procedure to <1%, as is the standard in the field [93]. The FACS sorting for primitive HSPCs further enhanced the obtained purity substantially. The consecutive lineage depletion including CD3^+^ T cells (>99% purity), gating out Lin^+^ cells, and sorting for stem/progenitor cells with a sort impurity of 1–3% rendered the likeliness for T cells extremely small. For example, a sort of 1000 HSCs with a high sort impurity of 3% would yield 30 cells off target. When considering 1% of these cells as T cells, statistically, in every third sort there would be one T cell. Furthermore, since T cells only comprise less than 5% of bone marrow mononuclear cells, these odds were even lower.

PM1 cells were obtained from the NIH AIDS Research Program (Germantown, MD, USA) and ACH-2 cells were kindly provided by Hans-Georg Kräusslich (Department of Infectious Diseases, Center for Integrative Infectious Disease Research (CIID), Heidelberg, Germany) and Frank Kirchhoff (Institute of Molecular Virology, Ulm University Medical Center, Ulm, Germany). Both cell lines were cultured in complete RPMI medium, composed of RPMI 1640 medium supplemented with 10% fetal calf serum (FCS, Pan Biotech, Aidenbach, Germany), 4 mM l-Glutamine, and (100 units/mL)/(100 µg/mL) Pen/Strep. HEK 293T cells and TZM-bl cells were obtained from the NIH AIDS Reagent Program and grown in Dulbecco’s Modified Eagle Medium (DMEM, Thermo Fisher Scientific) supplemented with 10% FCS, 4 mM l-Glutamine, and (100 units/mL)/(100 µg/mL) Pen/Strep.

### 2.3. Antibodies Used for Flow Cytometry

Antibodies to the following human proteins were used for flow cytometry: CD4 (clone RPA-T4, BrilliantBlue 515-conjugated, BD Bioscience, Franklin Lakes, NJ, USA), CD34 (clone 8G12, allophycocyanin (APC)-conjugated, BD Bioscience), CD389 (clone HB7, V450-conjugated, BD Bioscience), CD45RA (clone HI100, phycoerythrin (PE)-Cyanin7 (Cy7)-conjugated, Thermo Fisher Scientific), CD90 (Thy-1) (clone eBio5E10, PerCP-Cy5.5-conjugated, Thermo Fisher Scientific), CD184 (CXCR4, clone 12G5, PE-conjugated, Miltenyi Biotec), CD195 (CCR5, clone REA245, PE-conjugated, Miltenyi Biotec), CD3 (clone BW264/56, APC-conjugated, Miltenyi Biotec), and HIV-1 core antigen (p24) (clone KC-57, RD1-conjugated, Beckman Coulter). The secondary reagent used was streptavidin (APC-eFlour^®^780-conjugated, Thermo Fisher Scientific) to fluorescently label biotinylated lineage antibodies in the Lin-depletion kit in order to exclude remaining Lin^+^ cells via FACS. For HIV-1 p24 antigen staining, cells were fixed and permeabilized using the FIX & PERM^®^ kit according to the manufacturers’ instructions (Thermo Fisher Scientific). Samples were analyzed using a BD FACSCanto II after fixation (3% formaldehyde). Cell sorting was performed using a BD FACSAria Fusion (all BD Bioscience) flow cytometer. Compensation was performed using single-stained Lin^+^ cells for each marker.

### 2.4. Virus Preparation and Transductions

#### 2.4.1. Replication-Incompetent Pseudotyped Reporter Viruses

Replication-incompetent HxB2 X4 Env and JRFL R5 Env-pseudotyped reporter viruses were produced by transient transfection of HEK 293T cells using calcium phosphate precipitation as previously described [96]. Briefly, 1 µg of *env* expression plasmid (HxB2 (produced by and received from G. Melikyan) or JRFL (produced and donated by M. Dittmar)), 12.5 µg of *gag-pol* packaging plasmid (pCMV-dR8.91) [97], and 7.5 µg of GFP-encoding transfer plasmid (pHR’) [98] were co-transfected. The medium was exchanged 6 h post-transfection. Cell culture supernatants were collected 24 h and 48 h after transfection, concentrated via centrifugation o/n at 4 °C (6000 rpm/5810× *g,* Beckman Instruments, Brea, CA, USA), and stored at −80 °C.

#### 2.4.2. Replication-Competent Reporter Viruses

The molecular clone for the generation of replication-competent HxB2 reporter virus (pR7/3-EGFP) was kindly donated by O.T. Keppler (Max von Pettenkofer Institute, Munich, Germany). Virus was produced via transient transfection of 293T cells with 12.5 µg of pR7/3-EGFP using calcium phosphate precipitation. Culture supernatants were collected 24 h and 48 h post-transfection, filtered (0.45 µm), and stored at –80 °C.

#### 2.4.3. Replication-Competent Wild-Type HIV-1 Viruses

Replication-competent wild-type HIV-1 strains Lai and JRCSF were propagated in PM1 cells. Infectious supernatants were filtered (0.45 µm) and stored at −80 °C. Virus titers of replication-competent wild-type HIV-1 were determined using TZM-bl cells as previously described [99]. Infection of cells with replication-competent wild-type HIV-1 was performed for 48 h with defined multiplicities of infection (MOI), i.e., defined numbers of infectious viruses per cell.

#### 2.4.4. Virus Titer Determination

Virus titers of replication-incompetent and replication-competent reporter viruses were measured using serial dilutions of the virus stocks in single round infection assays in PM1 cells. Cells were transduced with different MOIs via spin inoculation (centrifugation at 2000 rpm/654× *g*, 90 min at 32 °C, Megafuge 1.0R, Thermo Fisher Scientific, or Rotanta 460R, Hettich Centrifuges, Tuttlingen, Germany). Transduction was quantified by measuring GFP-expressing cells via flow cytometry 72 h post-transduction, and virus titer was calculated considering the dilution and total cell number at the time point of transduction. Samples with less than 20% of GFP-expressing cells were used for the calculation to minimize bias based on multiple transductions per cell.

### 2.5. Inhibition Experiments

Lin^−^ cells were sorted for CD34^+^CD38^−^CD45RA^−^ cells and cultured overnight in SFEM medium. Prior to transduction, cells were incubated with the mouse monoclonal anti-CD4 QS4120 inhibitory antibody (10 µg/mL) or the small molecular drug and CXCR4 antagonist AMD3100 (1 µg/mL) (both EMD Millipore, Burlington, MA, USA) for 30 min. Cells were transduced without spin inoculation using replication-incompetent HxB2 Env-pseudotyped reporter virus (MOI = 50) and incubated for five days. At day six, cells were analyzed for GFP^+^ transduction events using flow cytometry.

### 2.6. Colony-Formation Assays

Colony-formation assays were performed using lineage-negative bone marrow cells isolated from PLHIV or HxB2-transduced (MOI = 5) Lin^−^ cells from cord blood or bone marrow samples of healthy donors. Colony assays were performed using the MACS^®^ Media HSC-CFU complete kit with Epo according to the manufacturer’s instructions (Miltenyi Biotech). Colonies were scored according to the color, size, and granularity of the cells within the colonies and checked for GFP^+^ expression after 14–16 days via light and fluorescence microscopy.

### 2.7. Replating Assay

Replating assays were performed using magnetically selected Lin^−^ cells transduced with replication-incompetent HxB2 Env-pseudotyped reporter virus (MOI = 5). After transduction, cells were grown in SFEM-complete medium for five days and subsequently sorted for GFP^+^ expression via flow cytometry. GFP^+^ cells were plated on 3-centimeter culture dishes in methylcellulose using MethoCult™ H4034 Optimum (STEMCELL Technologies). Five hundred GFP^+^Lin^−^ cells/plate were incubated for seven to eight days in the incubator at 37 °C and 5% CO_2_. Grown colonies were checked for GFP^+^ expression via fluorescence microscopy. GFP^+^ colonies were picked (with the help of a binocular), scattered, reseeded in methylcellulose, and incubated for another seven to eight days at 37 °C and 5% CO_2_. This working cycle was repeated until no further apparent growth of GFP^+^ colonies could be detected.

### 2.8. Analysis of env Sequences from HIV+ Bone Marrow

CD34^+^, Lin^−^CD34^−^, and Lin^+^ (control) populations were magnetically isolated from bone marrow samples of six PLHIV using the CD34 MicroBead Kit/human according to the manufacturer’s instructions (Miltenyi Biotec). Bone marrow mononuclear cells were pre-incubated with the biotinylated antibody cocktail from the Lineage Cell Depletion Kit, human (Miltenyi Biotec) for 10 min at 4 °C. After washing the cells with MACS-buffer, they were labeled using streptavidin (APC-eFlour^®^780 conjugated, Thermo Fisher Scientific) and α-CD34 (clone 8G12, allophycocyanin (APC) conjugated, BD Bioscience) antibody. The cell populations were sorted using flow cytometry (BD FACSAria™ Fusion). The total DNA was isolated using the QIAamp^®^ DNA Blood Mini Kit according to the manufacturer’s instructions (Qiagen, Hilden, Germany). Nested PCRs of *env* constructs encompassing gp120 and parts of gp41 (HxB2 location 6202–7838) were performed as follows: first-round PCR with primers EnvA (5′-GGC TTA GGC ATC TCC TAT GGC AGG AAG AA-3′; corresponding to nucleotides (nt) 5954 to 5982 of the HxB2 sequence) and gp120out (5′-GCA RCC CCA AAK YCC TAG G-3′) using PrimeSTAR GXL DNA polymerase (Takara Bio Inc, Kasatsu, Shiga, Japan), second-round PCR with primers EnvB (5′-AGAAAGAGCAGAAGACAGTGGCA-3′) and gp120in (5′-CGT CAG CGT YAT TGA CGC YGC-3′) using Platinum Taq polymerase (Life Technologies, Carlsbad, CA, USA), both according to the manufacturer’s instructions. The bulk PCR products were directly sequenced and additionally cloned into a pCR4 TOPO vector and transformed into MAX Efficiency™ Stbl2™ competent cells according to the manufacturer’s instructions (Life Technologies, Carlsbad, CA, USA). Transformed *E. coli* colonies were identified via colony PCRs using DreamTaq Green PCR Master Mix (2×) with universal vector-specific primers M13F/R (pCR4 TOPO) according to the manufacturer’s instructions (Thermo Fisher Scientific). Positive clones were cultured in LB_Amp_ medium overnight and their plasmids were isolated using the peqGOLD Plasmid Miniprep Kit I according to the manufacturer’s instructions (VWR International, Radnor, PN, USA). Gp120 *env* sequences were determined via sequencing using universal primers M13F/R (pCR4 TOPO). The sequencing was performed by GATC Biotech (Konstanz, Germany). Sequence analysis was performed using DNASTAR Software (Madison, WI, USA) and MEGA v.5.2 Software (https://www.megasoftware.net/; accessed on 7 November 2021; Pennsylvania State University, PN, USA).

### 2.9. qPCR-Assay

CD3^+^ cells were isolated from bone marrow samples of PLHIV using the human CD3 MicroBeads Kit and the flow-through was used to isolate CD34^+^ cells using the human CD34 MicroBead Kit, both according to the manufacturer’s instructions (Miltenyi Biotec). The genomic DNA was isolated using the QIAamp^®^ DNA Blood Mini Kit (Qiagen) according to the manufacturer’s instructions, including two sequential elution steps. The genomic DNA of the Jurkat and PM1 T cell lines, as well as genomic DNA of mononuclear cells of two cord blood donors, was isolated as the negative control. Quantitative PCR was performed as described previously [100] using forward primer M667 (5′-CTA ACT AGG GAA CCC ACT GCT TAA-3′), reverse primer M661 (5′-TCG AGA GAT CTC CTC TGG CTT-3′), and FAM-labeled probe M667 (FAM-TTG AGT GCT CAA AGT AGT GTG TGC CCG TCT GTT-TAMRA). The total HIV-1 DNA, including genomic and episomal DNA, was detected [101]. In each experiment, the housekeeping gene ribonuclease P (RNase P) served as the amplification control, and DNA levels were normalized to ng DNA.

### 2.10. Confocal Imaging

HSPC subsets were sorted via flow cytometry as described above and transduced using replication-incompetent HxB2 Env-pseudotyped reporter virus (MOI = 10). Sorted cells, transduced or untransduced, were stained for confocal imaging after incubation for three days at standard conditions. For confocal imaging, cells were pipetted onto Superfrost^®^ plus slides (Thermo Fisher Scientific). After a short period of drying, cells were then fixed and permeabilized using the FIX & PERM^®^ Kit according to the manufacturers’ instructions (Thermo Fisher Scientific). The permeabilized cells were washed carefully using wash buffer (PBS supplemented with 3% FCS) and stained using 0.2 µg of α-CXCR4 12G5 [102] in combination with 1 µg of goat α-mouse AF546 (Thermo Fisher Scientific) and 1 µg of α-CD4 OKT4 AF647 (BioLegend, San Diego, CA, USA) diluted in 200 µL PBS as well as DAPI. The primary antibody was diluted in wash buffer and incubated for 1.5 h at room temperature. The secondary antibody was incubated for 30 min at room temperature. The α-CD4 antibody was also diluted in wash buffer and incubated at room temperature for 1.5 h. The cells were washed carefully between staining steps using wash buffer. After completion of the antibody staining, 1 µg/mL DAPI was added to the cells, which were then incubated for 15 min at 4 °C in the dark. The cells were washed three times using PBS before 15 µL of ProLong™ Gold Antifade Mountant (Thermo Fisher Scientific) were applied and the coverslip was added. The slides were incubated overnight at 4 °C. Confocal images were taken using a confocal laser scanning microscope (Leica Microsystems, Wetzlar, Germany).

### 2.11. Correlation Analysis

Spearman rank correlation matrices were created using GraphPad Prism version 8.4.3 (GraphPad, San Diego, CA, USA). Correlograms were generated using the corrplot package in program R v.4.1.0 and R Studio v.1.4.1106 [103,104].

With the given sample size of 11 and an error of 5%, 80% power was achieved at an upper and lower critical r value of +/− 0.60 (G*Power v.3.1.9.4). *p*-values < 0.05 were considered significant; significance values are indicated as * *p* < 0.05, ** *p* < 0.01, *** *p* < 0.005.

## 3. Results

### 3.1. In Vitro Infection/Transduction of Cord Blood and Bone Marrow-Derived HSPCs with HIV-1

To analyze the susceptibility of lineage-marker-negative (Lin^−^) progenitor cells from cord blood and bone marrow to HIV-1 entry in vitro, we applied a stringent two-round lineage depletion protocol (Appendix A). It reduced the likeliness of CD3^+^ T cell contaminants in the Lin^−^ population to <1%, comparable to the standard in the field [93]. The sorted Lin^−^ progenitor cells were transduced 24 h post-selection using replication-incompetent reporter viruses encoding for GFP and pseudotyped with HIV-1 HxB2 (X4) or HIV-1 JRFL (R5) Env. GFP expression was analyzed 72 h post-transduction using flow cytometry and 5 days post transduction using fluorescence microscopy, the latter to maximize GFP signals (Figure 1a,b, Appendix A).

While Lin^−^ cells could not be transduced with R5 reporter virus, GFP^+^ transduction events were recorded with X4 reporter virus in Lin^−^ cells obtained from both cord blood and bone marrow. Pretreatment using the reverse transcriptase inhibitor efavirenz (EFV) abrogated GFP expression, which excluded GFP protein transfer as a source of the signal. Generally, increased transduction efficiencies were obtained with a higher multiplicity of infection (MOI). Significant transduction could be detected for bone marrow-derived Lin^−^ cells at an MOI of 1 and for cord-blood-derived Lin^−^ cells at an MOI of 10 (Figure 1a).

HIV-1 entry into HSPCs was verified using replication-competent HIV-1 (Figure 1c–e). First, the CXCR4-tropic HxB2 virus was used, in which the *nef* cassette was exchanged for eGFP to allow visibility and quantification of infection events (Figure 1c and Appendix A). Second, to exclude infection and detection artifacts that might appear with the usage of GFP reporter viruses, we performed infection experiments using wild-type X4 Lai and R5 JRCSF viruses, in which infection was quantified by intracellular p24 capsid (p24) staining after 48 h (Figure 1d and Appendix A) [105]. In accordance with the data obtained with pseudotyped reporter viruses, Lin^−^ cells derived from both cord blood and bone marrow were sensitive to X4 HIV-1 infection. Infection events were recorded at an MOI of 1 for bone marrow-derived HSPCs and an MOI of 3.5 for cord-blood-derived HSPCs (Figure 1c). Interestingly, infection of Lin^−^ cells was also observed with R5 (JRCSF) wild-type virus, although p24 levels were much lower compared to X4 (Lai) infection (Figure 1d). A summary of reporter virus constructs used in this study is shown in Figure 1f–i.

Given the preferential transduction of HSPCs by X4 viruses, we analyzed the dependence of entry on the presence of CD4 and/or CXCR4 receptors in HxB2 pseudotyped reporter virus transduction experiments in FACS-sorted, very immature HSCs and MPPs (CD34^+^CD38^−^CD45RA^−^ Lin^−^) in the presence of specific inhibitors. Blocking experiments were performed with cord-blood-derived FACS-sorted CD34^+^CD38^−^CD45RA^−^Lin^−^ cells using an inhibitory α-CD4 antibody and the CXCR4 antagonist AMD3100, which were added 30 min prior to transduction with HxB2 pseudotyped virus (MOI = 50) (Figure 1e and Appendix A). Both inhibitors resulted in a reduced number of GFP-positive cells compared to untreated controls, supporting the functional implication of both receptors in HSPC infection (α-CD4: 3.4-fold; AMD3100: 4.5-fold reduction).

### 3.2. Co-Expression of CD4 and HIV Co-Receptors on HSPC Subsets

Based on the functional importance of CD4 and CXCR4 receptors for HSPC transduction, we investigated the frequency of CD4/CXCR4 and CD4/CCR5 double-positive cells in cord blood and bone marrow HSPC subsets from nine donors to define subpopulations potentially susceptible to HIV-1 infection. Magnetically sorted Lin^−^ cells were co-stained for early HSPC markers (CD34, CD38, CD45RA, and CD90) as well as HIV-1 receptors (CD4, CCR5, and CXCR4), and analyzed using flow cytometry (representative FACS plots are shown in Appendix A). CD4 and CXCR4 co-expression was detected at considerable levels in all subsets analyzed (range of mean values: 0.55–5.90%). CD4 and CCR5 co-expression (range of mean values: 0.02–4.14%) was consistently and significantly lower in all sorted, primitive subsets. Apart from the entirety of Lin^−^ cells (>4% CD4^+^/CXCR4^+^ cells in cord blood and bone marrow), the number of CD4^+^/CXCR4^+^ cells was highest in cord blood and bone marrow-derived HSCs (CD34^+^CD38^−^CD45RA^−^CD90^+^ Lin^−^; cord blood mean = 4.53%, bone marrow mean = 4.15%). Bone marrow-derived MPPs (CD34^+^CD38^−^CD45RA^−^CD90^−^) exhibited the lowest frequency of CD4^+^/CXCR4^+^ cells (mean = 0.55%), which was also significantly lower compared to MPPs from cord blood (mean = 2.06%). Lineage-committed progenitors (CD34^+^CD38^+^Lin^−^) from cord blood and bone marrow had comparable frequencies of CD4/CXCR4 double-positive cells (mean values of 1.42% and 1.53%, respectively). The frequency of CD4^+^/CCR5^+^ cells was significantly higher in bone marrow (mean value of 4.13%) compared to cord blood (mean value of 0.75%) Lin^−^ cells. The sorted, primitive HSPC subsets demonstrated comparably low levels (mean values mostly <1%) of CD4^+^/CCR5^+^ cells in cord blood and bone marrow, whereby the abundance was lowest in bone marrow-derived MPPs (mean value of 0.02%, Appendix A).

### 3.3. HIV (Co-)Receptor Expression and Localization on HSPCs Using Confocal Micrographs

In addition to flow cytometry, confocal microscopy was applied to study HIV (co)receptor expression and localization, both before and after the transduction of CD34^+^CD38^+^ progenitors, MPPs, and HSCs sorted from cord blood. The co-expression of CD4 and CXCR4 on the cell surface of sorted CD34^+^CD38^+^ progenitors, MPPs, and HSCs could be confirmed on confocal micrographs at the single-cell level (Appendix A). After transduction with replication-incompetent HxB2 pseudotyped reporter virus, GFP^+^ cells exhibited enhanced amounts of internalized CD4 and CXCR4 in the cytoplasm compared to GFP^−^ cells. (Appendix A).

### 3.4. Susceptibility of Defined HSPC Subsets to X4 and R5-Mediated HIV-1 Entry

After demonstrating the co-expression of CD4/CXCR4 and CD4/CCR5 in the studied HSPC subsets, we analyzed the susceptibility of purified early HSPC subsets to HIV-1 Env-pseudotyped reporter virus transduction. Bone marrow-derived Lin^−^ cells were sorted by FACS into defined HSPC subsets, i.e., CD34^+^CD38^+^ progenitors, MPP, and HSC, according to the markers indicated above. The FACS-sorted cells were transduced using different MOIs of HxB2 (X4) or JRFL (R5)-pseudotyped reporter viruses and analyzed for GFP expression three days after transduction (Figure 2 and Appendix A). When using HxB2-pseudotyped reporter virus, positive transduction events could be detected in all HSPC-subsets, including the most naïve HSCs, which is in line with the previous co-receptor analyses. GFP^+^ events increased with higher MOI, as shown for CD34^+^CD38^+^ progenitors (MOI = 1:0.22%; MOI = 10:0.77%) and for MPPs (MOI = 1:0.28%; MOI = 10:0.79%). However, the observed MOI-dependent difference was only significant for CD34^+^CD38^+^ progenitors, not for MPPs (Figure 2 and Appendix A).

When using R5-pseudotyped reporter virus, GFP^+^ transduction events were observed exclusively in CD34^+^ CD38^+^ progenitors in an MOI-dependent manner (MOI = 1:0.012%; MOI = 10:0.063%). The R5 transduction rates at MOI = 10 were significantly lower compared to X4 (Appendix A), following the trend seen in Lin^−^ cells (Figure 1). MPPs and HSCs were not susceptible to JRFL (R5)-pseudotyped reporter viruses.

### 3.5. Multi-Lineage Colony Formation Capacity of HIV-1-Transduced Hematopoietic Progenitors

To prove HIV-1 entry into multipotent HSPCs functionally, we analyzed the capacity of multi-lineage colony formation of Lin^−^ cells after transduction with replication-incompetent HxB2-pseudotyped reporter virus. Transduced bone marrow- and cord-blood-derived cells were sorted for GFP expression five days after transduction and seeded in methylcellulose to perform colony-forming assays (Figure 3a,b). The transduced cells were able to form different colony types, including granulocyte/macrophage (GM) and granulocyte/erythrocyte/macrophage/megakaryocyte (GEMM) colonies, indicating that multipotent and/or functionally different oligopotent progenitors were transduced. Of note, a lower amount of erythrocyte (E) colonies was detected among GFP^+^ colonies (GFP^+^: 51.12%, GFP^−^: 65.53%), while the amount of mixed GM colonies was significantly higher in GFP^+^ colonies compared to GFP^−^ (GFP^+^: 20.69%, GFP^−^: 8.45%; *p* = 0.0218, two-tailed, unpaired *t*-test, Figure 3b).

### 3.6. Long-Term Colony-Forming Capacity of HIV-1-Susceptible HSPCs

Previous experiments have demonstrated that primitive HSPCs are susceptible to HIV-1 transduction and are able to differentiate into colonies of different lineages. It remained elusive whether HIV-1 Env-pseudotyped reporter-virus-transduced progenitor cells could retain their colony-forming potential over more extended periods and multiple cycles. To address this question, replating assays were performed using cord-blood- and bone marrow-derived Lin^−^ cells. Magnetically sorted Lin^−^ cells were transduced with HxB2 (X4)-pseudotyped reporter virus and sorted for GFP expression via FACS five days after transduction. GFP^+^ cells were seeded into growth factor-replete methylcellulose to promote differentiation and colony growth. After 7–8 days, GFP^+^ colonies were picked, resuspended, and seeded again in methylcellulose for 7–8 days. This replating was repeated until the colony growth of GFP-expressing cells ceased (Figure 4a). When using bone marrow-derived cells, growth of GFP-expressing colonies could be detected for up to three replating steps for all donors screened, resulting in a mean of 26.8 cultivation days (26 days, *n* = 3; 29 days, *n* = 1). Notably, cord-blood-derived cells maintained GFP^+^ colony growth for up to six replating steps and up to 48 days. A mean of 34 days of cultivation was achieved for all donors analyzed (*n* = 4), underscoring the functionality of HIV-1 infected progenitors (Figure 4b).

### 3.7. Colony Formation of Bone Marrow-Derived Lin^−^ Cells from HIV^+^ Individuals

As HIV-1 infection has been linked to multiple hematopoietic defects in vivo [61,62,63], we addressed the question of whether HSPCs derived from PLHIV are functionally impaired regarding colony formation. For this purpose, we compared the colony-forming capacity of magnetically sorted bone marrow-derived Lin^−^ cells isolated from four PLHIV and three healthy controls (Figure 5). In each of the five experiments, the total colony forming units (CFU) were reduced in Lin^−^ cells from HIV^+^ individuals, resulting in a significant overall reduction of 3.2-fold (*p* = 0.0194) when compared to healthy individuals and normalized to colony growth per 100 Lin^−^ cells. A reduction of colony growth in PLHIV was observed for each colony type. Interestingly, HIV-1 infection also skewed the proportions of colony types. In comparison to healthy donors, the percentage of granulocyte (G) and GM colonies was significantly higher/relatively better preserved among Lin^−^ cells from PLHIV. In contrast, the relative frequency of erythrocyte (E + EB), macrophage (M), and mixed GEMM colonies was slightly lower in PLHIV.

### 3.8. Clinical Evidence for HIV-1-Infected HSPCs in Bone Marrow from HIV-1-Infected Individuals

Bone marrow samples from eleven PLHIV were analyzed for HIV-1 proviral DNA in HSPCs using a highly sensitive qPCR assay amplifying a fragment of the HIV-1 R-U5 and *gag* region that is reverse transcribed late in the HIV-1 replication cycle (Figure 6 and Appendix A) [100]. Total DNA was isolated from CD34^+^ cells and CD3^+^ T cells (controls) of eleven donors, purified via double MACS isolation. After CD3^+^ T cell depletion, CD34^+^ purification was performed using two columns sequentially, i.e., applying the flow-through of the first column onto the second column to obtain CD34^+^ cell purities with less than 0.5% CD3^+^ cell contamination [93]. The cell numbers obtained and used for qPCR analysis were within the range of 3.4 × 10^4^–3.7 × 10^6^ for CD3^+^ cells and 1.5 × 10^4^–1.2 × 10^6^ for CD34^+^ cells (Appendix A). As negative controls, we used DNA from cord-blood-derived mononuclear cells as well as Jurkat and PM1 cells. HIV-1 cDNA copies were detected in CD3^+^ cells from eight donors. In CD34^+^ cells, HIV-1 sequences could be detected in genomic DNA from eight of the eleven donors, six being positive in both CD3^+^ and CD34^+^ cells (Figure 6). Notably, five of these donors had significantly higher copy levels of late HIV-1 cDNA/ng within CD34^+^ cells compared to the respective CD3^+^ cells (*p* ≤ 0.005). These donors had been diagnosed with HIV-1 infection more than three years prior to sample acquisition. One donor exhibited detectable HIV-1 cDNA copies in neither CD3^+^ nor CD34^+^ cells. Genetic co-receptor tropism analysis of the infecting viruses was obtained for seven of the eleven studied individuals. Of interest, six out of eight individuals with detectable HIV late cDNA copies in their CD34^+^ cells exclusively demonstrated R5 sequences in genotypic tropism testing performed at the time of bone marrow aspiration. No X4 tropism was detected (the remaining two could not be determined). Correlation analysis of qPCR results (detection of HIV-1 cDNA in CD34^+^ and CD3^+^ cells, viral copy numbers, and cell counts) with clinical (CD4 count, CD4/CD8, time since diagnosis, and viral load) and personal (age and sex) data did not yield significant associations between HIV copy numbers in CD34^+^ cells, the number of analyzed CD34^+^ cells, or clinical or personal variables (Appendix A).

We further analyzed purified bone marrow samples from six additional PLHIV (Figure 7a) for the presence of proviral DNA using a nested PCR amplifying a fragment (~1.6 kb) of the HIV-1 envelope gene (*env*). The high sensitivity of the *env* PCR was confirmed in an endpoint dilution assay using ACH-2 cells that carried single, integrated proviral copies of the X4 HIV-1 Lai genome. The sensitivity of the PCRs on DNA extracted from ACH-2 cells, which were serially diluted in non-infected PM1 cells, allowed the detection of a single HIV-1 copy (i.e., DNA prepared from a single ACH-2 cell diluted in 10^5^ PM1 cells) in two of three samples (Appendix A). The same PCR was applied to DNA isolated from primary bone marrow HSPCs of the six PLHIV: four out of six individuals had detectable viral loads in plasma, including two who were therapy-naïve at the time of sample collection. Five out of six individuals had CD4 counts <500/µL, resulting in treatment recommendations according to the “German/Austrian treatment guidelines” (https://daignet.de/site-content/hiv-therapie; accessed on 5 July 2021). FACS-sorted CD34^+^, Lin^−^CD34^−^, and Lin^+^ (positive control) cells were screened for HIV-1-specific *env* sequences. Within the Lin^+^ subpopulation, *env* sequences could be detected in five out of six donors (Figure 7b). It was not possible to amplify HIV-1-specific *env* sequences within sorted CD34^+^ cells of the participants; however, one donor (BM023) had detectable *env* sequences within the Lin^−^CD34^−^ subpopulation. For BM023, bulk sequencing and cloning were performed. The phylogenetic clustering of the BM023 sequences with subtype B reference sequences was supported by high bootstrap values (>80), which confirmed subtype B virus infection of BM023 (Figure 7c). Geno2Pheno analysis of the BM023 V3 sequences (10 clones and consensus sequence) revealed false positive rates (FPR) of 17.9%, which, according to the “recommendations for determination of co-receptor usage” as part of the “German/Austrian treatment guidelines” (FPR < 5%: CXCR4 usage; FPR >15%: CCR5 usage), suggests that progenitor cells from BM023 were infected with an R5 subtype B virus (Figure 7d). Among the six analyzed HIV^+^ individuals, the cART-naïve subject BM023 had the highest age, lowest CD4/CD8 ratios, and the most recent date of HIV diagnosis (Figure 7a). The numbers of sorted and analyzed cells were comparable/in the same logarithmic range for BM023 (74,995 Lin^+^ cells, 64,170 Lin^−^ cells and 2138 CD34^+^ cells) and the other participants (e.g., in HIV-negative BM020: 46,250 Lin^+^ cells, 57,851 Lin^−^ cells, and 5190 CD34^+^ cells).

## 4. Discussion

Although HIV-1 infection of HSPCs has been discussed for several years, it is still unclear whether HSPCs play a clinically relevant role as a viral reservoir in PLHIV. Previously, it has been reported that HSPCs are preferentially infected/transduced by X4 HIV-1 strains in vitro, whereas both X4 and R5 sequences have been isolated from HSPCs in vivo [75,93]. Here, we confirm that different subsets of HSPCs, including stem cells, are susceptible to HIV-1 in vitro (Figure 8).

In particular, X4 HIV-1 strains were able to infect HSPCs consistently depending on cell-surface-expressed CD4 and CXCR4 as well as inoculating viral dose. In contrast, in vitro infection with R5 HIV-1 was only achieved in CD34^+^CD38^+^ progenitors and to a significantly lower extent compared to X4. Increased sensitivity of HSPCs to X4 infection in vitro has been reported [76]. Here, we further delineated the susceptibility of defined HSPC subsets (CD34^+^CD38^+^ progenitors, MPPs, and HSCs) to X4 HIV-1 infection. Moreover, the finding that less primitive, i.e., CD34^+^CD38^+^ progenitors, are susceptible to R5 HIV-1 as well indicates that HSPCs can be targeted by HIV-1 in every infected patient, irrespective of the tropism of the infecting strain. CD4/CXCR4 double-positive cells were present in every HSPC subset, being highest in primitive HSCs. Moreover, the frequency of CD4/CXCR4 double-positive cells was higher in every subset compared to CD4/CCR5 double-positive cells. The largest difference was obtained for both cord-blood- and bone marrow-derived HSCs, for which the frequency of CD4/CXCR4 double-positive cells was 4.53% and 4.15%, compared to 0.68% and 0.90% CD4/CCR5 double-positive cells in the respective compartments (Appendix A). Differential expression of HIV-1 co-receptors is in line with the preferential entry of X4 HIV-1 into all HSPC subsets demonstrated here (Figure 2). These findings are following previous co-receptor expression-dependent HIV-1 entry analyses of HSPCs, which focused on the co-expression of the receptors on multipotent progenitors in one study, or individual co-receptor expression analyses on specific progenitor subsets in a second study [75,93]. Our data provide information on the co-expression of CD4 and HIV-1 co-receptors on HSPC subsets, which is in the range of 0.55% to 4.52% and parallels the rates of in vitro infection in our and others’ studies [75,79,93]. Thus, different HSPC subsets differ in their susceptibility to HIV-1 infection depending on their expression profile of HIV-1 receptors/co-receptors. Sebastian et al. also found that HSPCs harbor heterogenic subsets with respect to HIV-1 infection. The authors observed preferential infection of a CD4^high^ population within HSPCs in vitro, including MPPs [93]. Other studies described the essential function of CXCR4 in the trafficking of HSPC to and within the bone marrow [89,90]. Interestingly, an increased presence of CD34^+^ DNAM-1^bright^CXCR4^+^ cells was found in chronic viral infections, including HIV-1 infection, which may reflect chronic hyperactivation [106]. The biological role of the HIV-1-sensitive CXCR4/CD4 and CCR5/CD4 double-positive HSPC subpopulations remains elusive.

Expression levels of HIV-1 entry receptors and, consequently, infection/transduction rates of HSPCs were generally low. Previous reports suggested that in CD34^+^ cells, there might be a block in virus replication prior to integration, presumably caused by known or unknown restriction factors, which can be overcome by increasing the MOI [107]. Indeed, another study highlighted the presence and importance of APOBEC3G in CD34^+^ cells [108]. APOBEC3G was primarily identified in stimulated CD34^+^ cells, whereas the mRNA expression levels of APOBEC3G were rather low in unstimulated CD34^+^ cells. An additional study demonstrated the importance of interferon-stimulated genes in limiting the infection of CD34^+^ cells by HIV-1 [109].

HSPCs transduced with X4 HIV-1 were capable of multi-lineage differentiation, as shown by the formation of different types of colonies in CFU assays, including the multi-lineage CFU-GEMM colonies. These findings match previous reports, where multi-lineage differentiation of HSPCs was detected using a different type of replication-incompetent reporter virus [75]. We further found that transduced HSPCs were capable of long-term colony formation, indicating that HIV-1 targets primitive progenitor cells with at least transient self-renewing capacities. Persistent colony growth of transduced cells was detected for up to 48 days. In earlier studies, replating assays with uninfected HSPCs demonstrated persistent colony growth anywhere from 16 and up to 60 days or more [110]. Others reported that 50% of primary plated HSPCs had conserved clonogenic activity in repopulation assays after four weeks [111]. Furthermore, engraftment of HSPCs transduced with HIV-1-pseudotyped viruses was reported in irradiated mice [76]. The authors used a replication-incompetent HxB2 Env-pseudotyped reporter virus to transduce CD133^+^ cells, which were transplanted into sublethally irradiated mice. Multi-lineage engraftment of reporter-positive cells was detected in bone marrow, spleen, and peripheral blood cells after 18-26 weeks, suggesting the transduction of HSPCs capable of in vivo self-renewal and multi-lineage reconstitution. Although the applied assays in these studies differed in their experimental setup, our in vitro results support the hypothesis that HSPCs are capable of contributing to an HIV-1 reservoir.

In patients, HIV-1 infection has been linked to different hematopoietic defects [61,62,63]. While indirect causes, such as general immune hyperactivation or toxic effects by either secreted or shed HIV proteins, have been known for a long time [112,113,114], the role of direct infection of HSPCs has been confirmed only in recent years [75,76,79,93,94,95].

In our study, qPCR analysis revealed HIV-1 *gag*^+^ DNA in CD34^+^ cells from 8 of 11 samples (73%) from PLHIV (Figure 6). In five of these eight samples, the HIV-1 DNA levels were elevated compared to the respective CD3^+^ cell population. Cell numbers were within the range of 1 × 10^5^–4 × 10^6^ and 2 × 10^4^–7 × 10^5^ for CD3^+^ and CD34^+^ cells, respectively. Among the samples with HIV^+^ CD34^+^ cells, we obtained an average of 3.6 *gag*^+^ copies per million cells (0.00036%, range 0.04–29 copies per million cells). Other studies detected proviral DNA in HSPCs in 4 of 9 [75], 6 of 11 [80], and 9 of 9 study samples [92]. Although small in study size, proviral DNA in HSPCs was frequently detectable in all these studies. We used a similar CD34^+^ cell purification protocol as in recent studies of the Collins group, in which almost 60% of donors exhibited HIV-positive CD34^+^ cells [93,94,95]. These authors observed a mean frequency of 2.4 proviral copies per million CD34^+^ cells (0.0002%; range 0.8–18 copies per million cells), which is only 1.5 times lower and thus highly comparable to our data. Bordoni et al. compared treated and untreated patients, revealing higher infection rates of CD34^+^ HSPCs in treatment-naïve patients (cART^+^ 7 copies/1000 cells; cART^−^: 72 copies/1000 cells) [92]. This may explain the lower infection rates in some of the above studies, including ours, where samples from patients on cART were analyzed. Recent data from patients during long-term cART suggest that HIV-infected CD4^+^ T cell clones are smaller in size but can be more stable than uninfected clones [115]. A study of HIV-1 subtype C infected patients in Africa found a correlation between infected HSPCs in the bone marrow and higher rates of anemia [74]. McNamara et al. reported a correlation between HIV-1 detection in HSPCs and a recent diagnosis of HIV-1 infection [80]. Such correlations could not be found in our study (Appendix A). Infection of CD34^+^ cells only had a non-significant tendency to correlate with the number of CD34^+^ cells studied, age of the patient, time since diagnosis, and viral load.

When we screened for proviral *env* sequences in CD34^+^, Lin^−^CD34^−^, and Lin^+^ cells from six PLHIV using conserved nested PCR primer pairs [116], only one sample demonstrated a PCR signal in the Lin^−^CD34^−^ population, and no positive events in CD34^+^ cells were recorded. The scarce HIV-1 proviral detection among the HSPC subsets can presumably be attributed to the limited number of CD34^+^ cells available for proviral *env* sequencing. While the qPCR approach yielded HIV-1 positive results in 73% of CD34^+^ samples tested using 2 × 10^4^–7 × 10^5^ cells, the cell number for *env* sequencing was reduced by a factor of >10 (e.g., 5190 CD34^+^ cells for BM020 and 2138 CD34^+^ cells for BM023). These cell numbers might have been too low to come upon one or a few HIV-1 infected cells according to the qPCR estimate of only a small number of HIV-1 copies (~3.6) per million cells. Although the used primers have been established for bulk amplification of diverse HIV-1 subtypes and recombinant forms, it cannot be excluded that the sensitivity might have been compromised for the detection of non-B provirus compared to proviral subtype B sequences (such as in ACH-2 cells, Appendix A). All five participants with undetectable provirus in the progenitor subsets were infected with non-B subtypes. Nevertheless, previous studies also found at best minimal evidence of proviral HIV-1 DNA in CD34^+^ cells, and the viral reservoir in HSPCs has mostly been assigned only minor relevance [70,71,72,73,77,78]. Arguably, some earlier studies might not have achieved the necessary sensitivity to detect the latent reservoir in CD34^+^ cells or focused on CD34^+^CD4^−^ cells that do not harbor the adequate HIV-1 sensitive phenotype, based on the absence of CD4 [77,79,93].

Our *env* sequence analysis revealed the detection of a CCR5-tropic HIV-1 clade B provirus in progenitor cells of the one positive donor (Figure 7). In addition, tropism analysis in six of the eight donors harboring proviral HIV-1 DNA in CD34^+^ cells as detected by qPCR assay found infection with R5 HIV-1 (Figure 6 and Appendix A). These findings were somewhat unexpected, considering the preferential in vitro infection of HSPCs by X4 strains in our studies as well as others. Nevertheless, these particular studies suggested that less primitive progenitors can also be targeted by R5 HIV-1 [76,79], and indeed, evidence for HSPC infection by R5 strains has been recently observed in vivo [93]. In that study, the authors also identified a higher number of R5 compared to X4 sequences in HSPCs of HIV-1-infected donors, suggesting that R5 infection of HSPCs can take place at similar or even higher frequencies compared to X4 infection. To add to this, CXCR4 tropism was predicted for 16 HIV-1 proviral amplicons isolated from HSPCs of 8 donors, while CCR5-tropic HIV-1 was predicted for 36 amplicons isolated from 17 donors. In agreement with the CCR5-tropic transduction of CD34^+^CD38^+^ progenitors in vitro in our study, both X4 and R5 HIV-1 appear to target HSPCs, whereby R5 infection/transduction is restricted to less immature progenitors. Furthermore, wild-type R5-tropic virus can infect HSPCs more efficiently (Figure 1d and Appendix A) compared to non-replication-competent virus (Figure 1a and Appendix A), which is presumably due to the higher number of surface-expressed Env molecules on infectious viruses [117,118] that may tolerate lower amounts of receptors for entry. Accordingly, studies using humanized mice observed infection of CD34^+^CD38^+^ progenitors by HIV-1 strains of all tropisms, including CCR5-tropic strains, albeit at much lower infection rates than by CXCR4-tropic or dual-tropic strains [91].

Our functional ex vivo assays demonstrated impaired colony formation of Lin^−^ cells isolated from PLHIV compared to healthy individuals. This manifested as reduced myeloid colony numbers, both overall and within each analyzed colony type. Similar observations have been made before [119,120,121]. Previous studies reported that productively infected HSPCs die during in vitro cell culture after stimulation of proliferation or differentiation [75]. It is possible that in PLHIV, most of the infected HSPCs die during lineage development. However, impaired lineage development has also been linked to the depletion of CD34^+^CD38^−^ progenitors mediated by plasmacytoid dendritic cells (pDCs) in chronic HIV-1 infection [66]. The authors proposed that pDCs secrete cytokines that lead to the suppression and exhaustion of hematopoiesis despite the inhibitory impact of pDCs on viral replication in chronic HIV-1 infection. Therefore, the direct effects of HIV-1 as well as indirect effects on HSPCs may account for impaired lineage development and colony formation.

## 5. Conclusions

Within this study, we confirmed the existence of HSPC subpopulations that are susceptible to HIV-1 infection in vitro. Moreover, we found cell surface expression of HIV-1 entry receptors on several HSPC subsets, which renders them susceptible to HIV-1 infection, thus potentially contributing to a viral reservoir. Although the infection rates of different HSPC subsets were low (mostly < 1.5%), CD34^+^ cells in bone marrow samples of PLHIV were a frequent (8/11 donors), but not consistent source of proviral HIV sequences in our study. It therefore remains elusive whether participants who tested negative for proviral HIV-1 in HSPCs exhibited a lower number of HIV-1 copies or HIV-1-infected HSPCs that remained below the detection threshold, or whether they were truly negative. Conclusions are limited due to the small patient cohort and moderate cell numbers available for this study. Furthermore, it is possible that rates of HIV proviral sequences in HSPCs may differ by subtype. Even though in vitro infection of HSPCs with different subtypes has been reported (reviewed in [93]), data regarding the potency of different HIV-1 subtypes to infect HSPCs in vivo remain incomplete. More research is needed on defined patient samples to reveal the mechanisms, i.e., the time point and consequences of HSPC infection by HIV-1 in vivo and to determine the extent to which infected HSPCs contribute to the formation of a viral reservoir in the human body.

## Figures and Tables

**Figure 1 cells-11-02968-f001:**
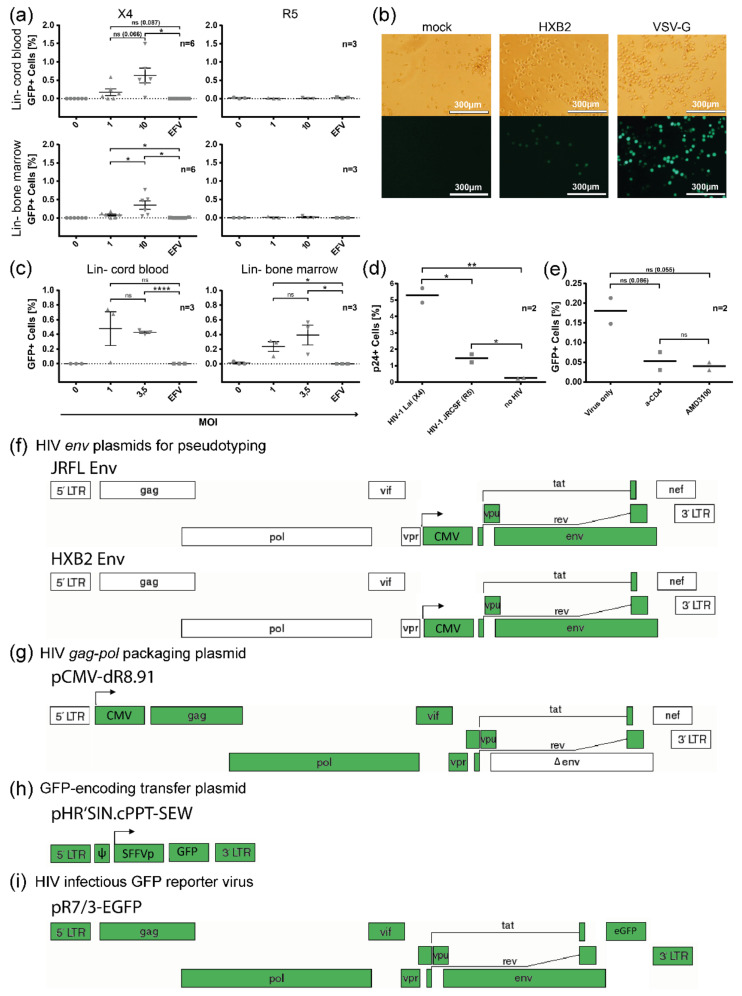
Susceptibility of cord blood and bone marrow-derived HSPCs to transduction with HIV-1 Env-pseudotyped reporter viruses as well as infection with replication-competent reporter virus strains and wildtype strains. (**a**) Transduction of cord blood and bone marrow-derived lineage-depleted (Lin^−^) cells using different MOIs of replication-incompetent HIV-1 Env-pseudotyped GFP reporter viruses. The percentage of GFP-positive cells is shown 72 h post-transduction with pseudotyped reporter virus encoding for CXCR4-tropic (X4) HxB2 Env (left) and CCR5-tropic (R5) JRFL Env (right). Abrogated transduction (MOI = 10) in cells treated with the reverse transcriptase inhibitor efavirenz (EFV) was used as negative control. (**b**) Fluorescence micrographs of cord-blood-derived Lin^−^ cells five days after transduction with reporter viruses pseudotyped with HxB2 or VSV-G Env (control) (MOI = 10). (**c**) Percentage of infection of cord-blood- and bone marrow-derived Lin^−^ cells using different MOIs of GFP-expressing replication-competent HxB2 (X4) reporter virus. (**d**) Percentage of p24^+^ cells 48 h post-infection of cord-blood-derived Lin^−^ cells using wild-type X4 Lai or R5 JRCSF HIV-1. (**e**) Inhibition of HxB2 Env-pseudotyped reporter virus transduction of cord-blood-derived CD34^+^CD38^−^CD45RA^−^ cells using α-CD4 antibody or CXCR4 antagonist AMD3100. Cells were lineage depleted and sorted via FACS. The mean and standard error of the mean are shown; n represents the number of different donors analyzed. In (**a**,**c**), all statistically significant differences using unpaired two-tailed *t*-test are shown (*: *p* < 0.05; **: *p* < 0.01; ****: *p* < 0.0001) and selected non-significant differences (n.s.) *p*-values are indicated for the range of >0.05 and <0.1. (**f**–**i**) Besides infectious, unmodified HIV virus, we used different infectious and pseudotyped reporter constructs. For pseudotyped viruses, three plasmids were used (**f**–**h**); the parts highlighted in green are included in the viral constructs.

**Figure 2 cells-11-02968-f002:**
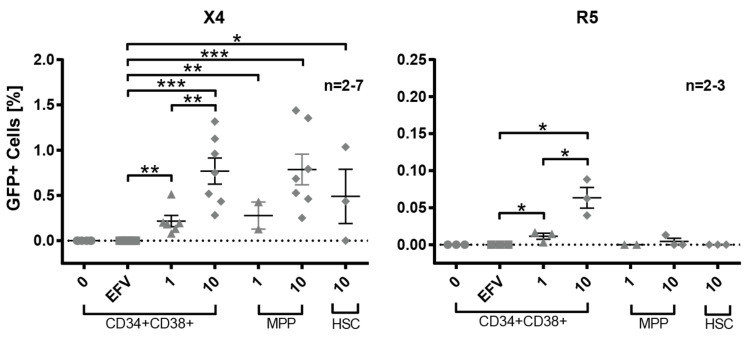
Transduction of bone marrow-derived sorted hematopoietic progenitor subsets with X4 and R5 HIV-1 Env-pseudotyped reporter viruses. Sorted hematopoietic progenitor subsets were transduced with GFP-expressing reporter viruses pseudotyped with X4 (HxB2, left) or R5 (JRFL, right) at different MOIs. The percentage of GFP^+^ cells 72 h post-transduction is shown. The sorted HSPC subsets are CD34^+^CD38^+^ progenitors, multipotent progenitors (MPP, CD34^+^CD38^−^CD45RA^−^CD90^−^) and hematopoietic stem cells (HSC, CD34^+^CD38^−^CD45RA^−^CD90^+^). Transduced cells (MOI = 10) treated with efavirenz (EFV) were used as negative control; n represents the number of different donors analyzed. The mean and standard error of the mean are indicated. Statistically significant differences to EFV-treated samples or among samples transduced with different MOIs per subset were calculated using a two-tailed *t*-test (* *p* < 0.05; ** *p* < 0.01; *** *p* < 0.001).

**Figure 3 cells-11-02968-f003:**
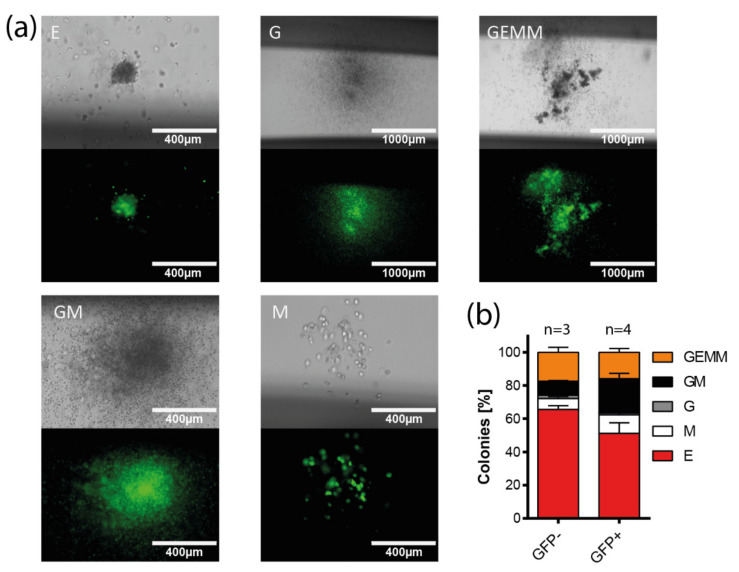
Distribution of HIV-1 infection in colonies formed from cord-blood-derived Lin^−^ cells. (**a**) Fluorescence images of GFP^+^ colonies obtained in a colony-formation assay (day 14), in which cord-blood-derived Lin^−^ cells were transduced with HxB2 Env-pseudotyped GFP reporter virus. (CFU: colony-forming unit; E: erythrocyte colony; G: granulocyte colony; mixed GEMM: granulocyte/erythrocyte/macrophage/megakaryocyte colony; GM: granulocyte/macrophage colony; M: macrophage colony) (**b**) Quantitative analysis of one colony-formation assay (d14), in which cord-blood-derived Lin^−^ cells were transduced with HxB2 Env-pseudotyped GFP reporter virus. The relative frequency of individual colony types is shown for transduced (GFP^+^) and non-transduced (GFP^−^) colonies. Two-tailed *t*-tests comparing GFP^−^ vs. GFP^+^ colony types show a significant difference for the amount of GM colonies (*p* = 0.0218). n represents the number of individual measurements; 413 GFP^−^ colonies and 74 GFP^+^ colonies were analyzed; mean and standard error of the mean are indicated.

**Figure 4 cells-11-02968-f004:**
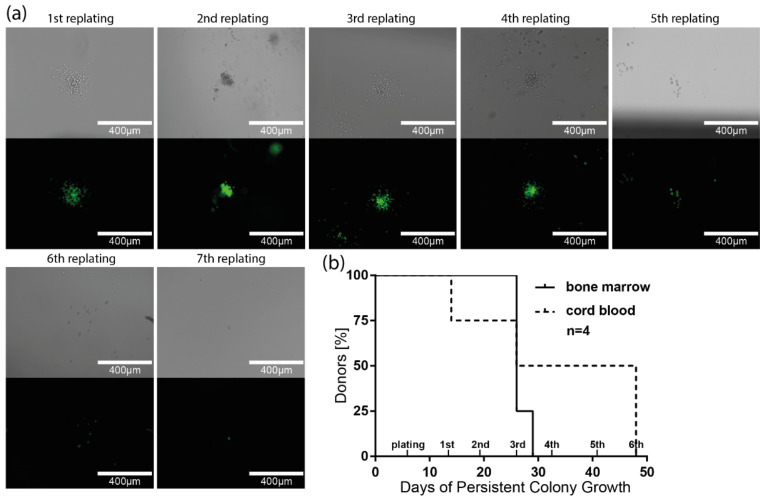
Replating assay and persistent colony formation of cord-blood- and bone marrow-derived Lin^−^ cells after transduction with HIV-1 Env-pseudotyped reporter virus. Micrographs of GFP^+^ colonies/cells after serial replating. Replating assays were initiated using Lin^−^ cells transduced with HxB2 Env-pseudotyped GFP reporter virus and sorted at day 5 for GFP^+^ expression. GFP^+^ cells were seeded in CFU medium and grown for 7–8 days before colonies were picked, scattered, and replated for another 7–8 days. Replating was repeated until GFP^+^ colony growth ceased. (**a**) Representative fluorescence micrographs of GFP^+^ colonies grown from cord-blood-derived progenitors after serial replating steps. First micrograph was taken after initial plating before first replating. (**b**) Percentage of donors with persistent colony growth of HIV-1 reporter-virus-transduced progenitors from cord blood and bone marrow over time; *n* represents the number of different donors analyzed for both cord blood and bone marrow.

**Figure 5 cells-11-02968-f005:**
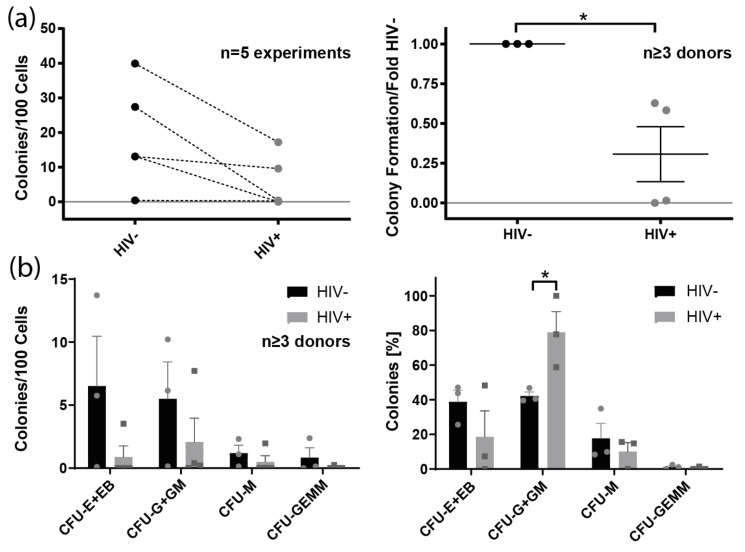
Impairment of colony formation of bone marrow-derived Lin^−^ cells of HIV-1 infected patients. Colony-formation assays of bone marrow-derived Lin^−^ cells from patients infected by HIV-1 and healthy controls. (**a**) Colony formation capacity of Lin^−^ cells from HIV-1 patients compared to uninfected donors showing the numbers of colonies per 100 cells seeded (**left**) and fold change normalized to uninfected controls within each assay (**right**); mean and standard error of the mean are shown for five experiments. (**b**) Effects of HIV-1 infection on the absolute colony numbers by subset formed per 100 cells seeded (**left**) and the relative colony numbers by subset (**right**). Mean, standard error of the mean, and individual data points are indicated. Two-tailed *t*-test comparing HIV^−^ vs. HIV^+^ groups: * *p* < 0.1. CFU: colony-forming unit; E: erythrocyte colonies; EB: erythroid burst colonies; G: granulocyte colonies; GEMM: mixed granulocyte/erythrocyte/macrophage/megakaryocyte colonies; GM: granulocyte/macrophage colonies; M: macrophage colonies.

**Figure 6 cells-11-02968-f006:**
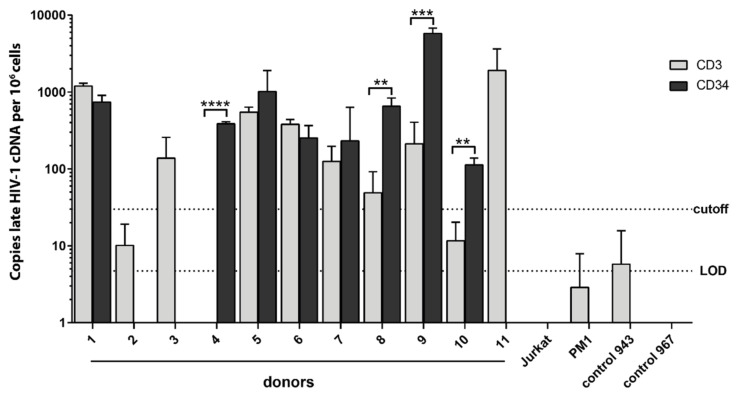
Detection of HIV-1 proviral DNA in CD34^+^ cells isolated from HIV-1 infected individuals. qPCR analysis using total (genomic and episomal) DNA of bone marrow-derived CD34^+^ and CD3^+^ cells isolated via sequential magnetic sorting from HIV-1 infected individuals. qPCR primers were specific for the HIV-1 R-U5 and *gag* region. Each sample was measured in technical triplicates. To determine the amount of copies per ng DNA, a PCR standard of the housekeeping gene RNAse-p was performed simultaneously on each sample. Numbers of late HIV-1 cDNA copies per ng gDNA (log_10_) are shown as mean and SD. HIV-1-negative cell-lines (PM1, Jurkat) and HIV-negative cord blood cells (control 943 and 967) were used as negative controls. Cut off was set to 3 × SD of control 943. LOD: limit of detection. ** *p* < 0.01; *** *p* < 0.001; **** *p* < 0.0001. Clinical data of study participants are shown in Appendix A.

**Figure 7 cells-11-02968-f007:**
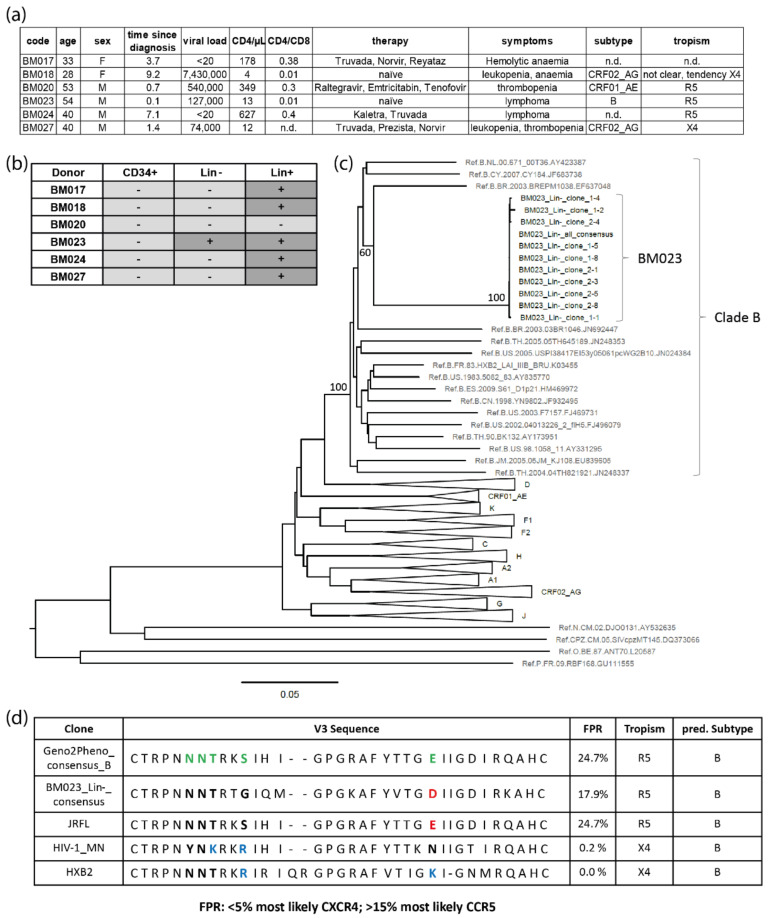
Screening for HIV-1 proviral DNA in hematopoietic progenitor cells from HIV-1-infected individuals. (**a**) Clinical data of screened donors. Units: age: age in years at bone marrow aspiration; time since diagnosis: years; viral load: RNA copies/mL; CD4: cells/µL; n.d.: not determined. (**b**) Detection of proviral HIV-1 DNA in sorted bone marrow-derived Lin^+^, Lin^−^, and CD34^+^ cells isolated from HIV-1-infected individuals using a nested PCR. Sensitivity of PCR: detection of one proviral genome in DNA of 10^5^ non-infected cells in 66% of all cases. (**c**) Neighbor-joining phylogenetic tree displaying *env* sequences (HIV region 6225-7817, HxB2 numbering), including the cluster of sequences isolated from hematopoietic Lin^−^ cells from bone marrow sample BM023. Reference sequences were downloaded from the Los Alamos National Library database. For clade B, reference sequences from all major branches are shown. Reference sequences from HIV-1 group M clades other than B are collapsed to triangles for simplicity. Bootstrap values are indicated for branches of major interest (calculated with 500 bootstrap replicates). The bar indicates a genetic distance of 5%. (**d**) Geno2Pheno analysis and interpretation: Consensus HIV-1 Envelope V3 loop sequence isolated from BM023 Lin^−^ cells compared to reference sequences and the consensus V3 loop sequence of HIV-1 clade B (Geno2Pheno; https://coreceptor.geno2pheno.org/index.php; accessed on 12 October 2021). Amino acids highlighted in green (consensus) are used to predict interaction with co-receptors; among these, negatively charged amino acids are highlighted in red in R5 sequences and positively charged amino acids are highlighted in blue in X4 sequences.

**Figure 8 cells-11-02968-f008:**
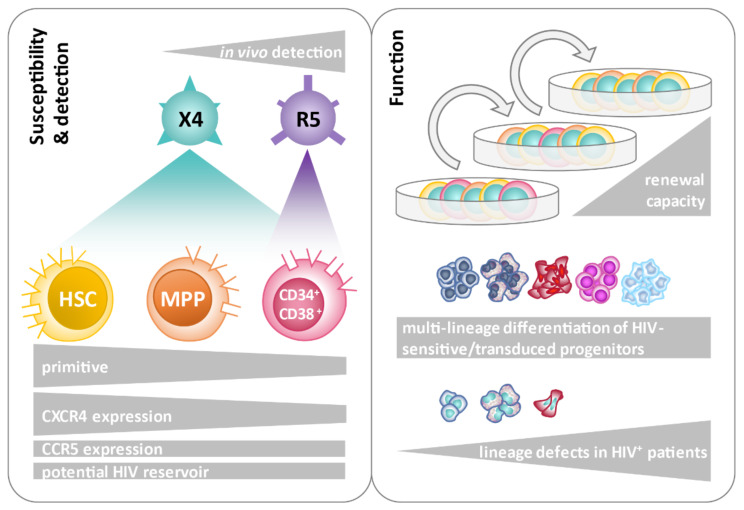
Schematic summary of the findings on the sensitivity of primitive hematopoietic progenitors to HIV-1 infection and functional consequences. The susceptibility of defined hematopoietic stem and progenitor cell subsets to CXCR4 (X4)- and CCR5 (R5)-mediated HIV-1 infection was determined in vitro and in vivo. The cell subsets were analyzed for cell surface expression levels of HIV entry receptors. Functional assays included serial replating of HIV-1-transduced progenitors and colony formation assays in differentiation-inducing medium to confirm the renewal and multi-lineage formation capacity of HIV-1 sensitive/transduced progenitors. A comparative analysis of colony formation between hematopoietic progenitors from HIV-1-infected and uninfected individuals unveiled lineage defects in HIV-1-infected samples. HSC: hematopoietic stem cells; MPP: multipotent progenitors.

## Data Availability

Primary data sets are available upon email request to the corresponding authors.

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
