# Peer review of "HIV-1 Infection of Long-Lived Hematopoietic Precursors In Vitro and In Vivo"

_cells, 2022, doi:10.3390/cells11192968_

Round 1

Reviewer 1 Report

Dear authors,

I read the article HIV-1 infection of long-lived hematopoietic precursors in vitro 2

and in vivo.

The subject is interesting and I have the following observations:

1. Please enter more details, comparisons with other studies, explanations in relation to your results in the discussion chapter.

2. Please correct the grammatical mistakes in English

3. Please double-check the references

Thank you

Author Response

Reviewer 1’s comments

  1. Please enter more details, comparisons with other studies, explanations in relation to your results in the discussion chapter.

Response:

  • We thank the Reviewer for the review of our paper and for providing feedback to improve the manuscript. According to the reviewer’s suggestion, we added further details to the Discussion and a new recent Reference about the stability of HIV-infected CD4+ T cell clones in treated HIV-infected patients. We added the new Figure 8 to the Discussion to convey the main message of the paper more clearly and to schematically summarize our main findings and conclusions. We also created a graphical abstract to explain the methodological proceeding and the key results.

  1. Please correct the grammatical mistakes in English

Response:

  • The new version of the manuscript was proofread by a native American English speaker, whom we acknowledged in the Acknowledgment section. All edits are included in “track changes” mode.

  1. Please double-check the references

Response:

  • We acknowledged the reviewer’s suggestion and removed former References 115 and 117 as they were only distantly related to the manuscript. Furthermore, we added a recent publication about the high stability of HIV-infected CD4+ T cell clones in patients under long-term antiretroviral therapy (new Reference 115).

Reviewer 2 Report

 The manuscript by Dr. Renelt et al. entitled "HIV-1 infection of long-lived hematopoietic precursors in vitro and in vivo" describes the possibility of HIV-1 infection in HSPCs subpopulations. The study evaluated several aspects of HIV-1 infection and reservation in evaluated cells by carrying out many suitable in vitro and in vivo experiments. The topic is exciting and has scientific merits.

In order to improve the manuscript, the reviewer hopes that the authors design a schematic figure of HSPCs infection with HIV-1, which explains underlying mechanisms and consequences.

Besides, A complete flowchart showing methods will obviously make the manuscript more comprehensive.

Author Response

Reviewer 2’s comments

  1. In order to improve the manuscript, the reviewer hopes that the authors design a schematic figure of HSPCs infection with HIV-1, which explains underlying mechanisms and consequences.

Response:

  • We thank the Reviewer for the useful idea. We added a schematic figure to the Discussion (Figure 8), which explains the underlying mechanisms and consequences.

  1. Besides, A complete flowchart showing methods will obviously make the manuscript more comprehensive.

Response:

  • We agree. We followed the Reviewer’s suggestion and added a flowchart showing the applied methods and associated findings as graphical abstract.